# Efficient Multi-Hop Wireless Power Transfer for the Indoor Environment

**DOI:** 10.3390/s23177367

**Published:** 2023-08-24

**Authors:** Janis Eidaks, Romans Kusnins, Ruslans Babajans, Darja Cirjulina, Janis Semenjako, Anna Litvinenko

**Affiliations:** 1Institute of Microwave Engineering and Electronics, Riga Technical University, Azenes St. 12, LV-1048 Riga, Latvia; janis.eidaks@rtu.lv (J.E.); romans.kusnins@rtu.lv (R.K.); darja.cirjulina@rtu.lv (D.C.); janis.semenjako@rtu.lv (J.S.); 2SpacESPro Lab, Riga Technical University, Azenes St. 12, LV-1048 Riga, Latvia; anna.litvinenko@rtu.lv

**Keywords:** wireless power transfer, multi-hop, internet of things, wireless senor networks, RF-DC conversion efficiency

## Abstract

With the rapid development of the Internet of Things (IoT) and wireless sensor networks (WSN), the modern world requires advanced solutions for the wireless powering of low-power autonomous devices. The present study addresses the wireless power transfer (WPT) efficiency problem by exploiting a multi-hop concept-based technique to increase the received power at the end sensor node (ESN). The current work adopts efficient multi-hop technology from the communications field to examine its impact on WPT performance. The investigation involves power transfer modeling and experimental measurements in a sub-GHz frequency range, chosen for being capable of providing a greater distance to transmit power. The paper proposes a multi-hop (MH) WPT concept based on signal amplification and demonstrates the fabricated multi-hop node (MHN) prototype. The experimental verification of the MHN is performed in the laboratory environment. The present paper examines two WPT scenarios: line-of-sight (LoS) and non-line-of-sight (NLoS). The turn-on angle of 90 degrees on MHN is used for the NLoS case. The received power and RF-DC converted voltage on the ESN are measured for all investigated scenarios. Moreover, the paper proposes an efficient simulation approach for the performance evaluation of MH WPT technology, providing an opportunity to analyze and optimize wireless sensor nodes’ spatial distribution to increase the received power.

## 1. Introduction

The growing number of smart interconnected devices and systems that make the Internet of Things (IoT) and the underlying wireless sensor networks (WSN) majorly impact various branches of the industry, creating smart interconnected environments with many low-power smart sensors that autonomously collect and exchange a huge amount of data [1,2]. The Ericsson Mobility Report [3] estimates cellular, wide-area, and short-range IoT connection to be 13.2 billion in 2022 and is expected to increase to 34.7 billion by 2028. The increasing integration of IoT is a significant driving force of economic development [4,5]. In 2013, the McKinsey Global Institute estimated the IoT economic impact to be between USD 2.7 and USD 6.2 trillion [6,7]; in 2021, the estimation was USD 5.5, being expected to rise to USD 12.6 trillion by 2030 [7]. The increasing number of autonomous devices and sensor nodes (SNs) poses a challenge to powering the devices employed in the network. The cases where the power delivery infrastructure cannot power an SN are usually covered by battery power. However, using batteries to power SNs poses a challenge in maintaining SNs by changing or charging batteries. The placement of the SNs can complicate maintenance or make it impossible. The increasing number of sensors and the need to maintain the integrity of the sensor network only amplifies the problem.

Radio frequency (RF) wireless power transfer (WPT) has developed over the last several years [8,9]. This technology not only benefits the powering solution but also facilitates control over the power consumption of an individual SN. WPT initially branched from the topic of RF energy harvesting (EH), which focused on powering individual autonomous SNs. The focus of RF EH was on developing solutions to convert the ambient RF signals to DC power. With the rapid increase in information exchange, RF EH approaches had to be complemented to be integrated into the WSN, thus moving from ambient EH to WPT. The simple structure of WPT thus consists of a power transmitter and a power receiver. The transmitter generates a power-carrying signal that the power receiver then converts to a DC voltage. The power receiver is an RF–DC converter paired with an antenna and an energy storage device (battery or a capacitor) connected to the SN. The main focus of current studies in the field is to enhance WPT system performance, with the most attention paid to the design of the rectenna (antenna paired with an RF–DC converter), studying possible RF–DC topologies, antenna designs, and their implementation [10,11,12,13].

The above-mentioned development of the WPT topic was single-device-scaled. An emerging concept of simultaneous wireless information and power transfer (SWIPT) has also received considerable attention, enabling backscatter technology [14,15,16]. The continuous development of WPT now shifts towards network-scale solutions, incorporating and combining new technologies to increase the performance of WPT on a network scale. The inspiration for such development is from the solutions developed for wireless communications.

In any wireless propagation, the wireless channel between the transmitter and receiver limits the system’s performance [17,18]. In the case of WPT, the goal is adaptive power-signal forming matched to the channel and RF–DC converter of the harvesting node with a goal to increase the DC converted power [19,20,21]. Utilizing the propagation environment in a more efficient way might result in better performance of the WPT system not only in the case of a single power transmitter and receiver but also on a network scale [22]. The work [23] demonstrates that theoretically, channel adaptive waveforms might increase the delivered DC power by 100%, and in [24], the experiments demonstrate that joint waveform and beamforming can boost output DC power by 100% and increase the WPT range. However, the design of the most efficient power-carrying waveforms may be computationally complex, and thus a compromise must be made between the achievable efficiency and application possibilities [23].

Progress in RF micro-electromechanical systems (MEMS), as well as the field of artificial materials, influenced a number of emerging transmission technologies, such as intelligent reflecting surfaces (IRSs) [25,26]. IRSs are a promising approach to enhancing the performance of wireless sensor networks, as they may, in the foreseeable future, replace their active counterparts, extensively employed in wireless communications. Recent studies have demonstrated that a further reduction in the energy consumption of IRS-assisted WSNs [27] as well as reduced transmit power in the case of satellite-terrestrial relay networking [28], can be achieved through passive beamforming optimization. This approach can be followed to achieve a higher SWIPT efficiency [29,30]. Beamforming itself is an essential extension of WPT aimed at increasing the power transfer range and steering the power-carrying signal direction to enhance performance [31]. Using IRSs for WPT can increase the distance of power transfer as well as improve the performance in the non-line-of-sight (NLoS) energy transfer case. The work [32] experimentally demonstrates a 20 dB gain in received power when utilizing IRS for WPT. However, for the optimal deployment of IRSs in WPT, the placement and number of surfaces must be considered [33].

Multi-hop routing is another promising technique that can enhance the power efficiency of wireless communication systems, WPT systems, or even SWIPT [34,35]. Applying this wireless communications technique to WPT simplifies the overall infrastructure by reducing the number of power transmitters (power beacons, PB). The multi-hop node (MHN) will have enough power for the connected SN and transmit some energy to another SN not covered by the PB. This approach can slightly extend the coverage of the WPT without extra PB and also improve the performance in the NLoS propagation.

The work [36] experimentally demonstrated that the voltage gain on the end node powered using a multi-hop approach could increase up to 32% depending on the configuration. However, unlike the energy-aware routing developed in conventional wireless networks, the routing protocols in multi-hop WPT systems must consider the RF energy propagation and the circuit design of network nodes. Thus, the power-efficient multi-hop nodes must be supplemented with power-efficient routing protocols [37,38]. This limits the potential scale of the deployed multi-hop WPT system.

Each of the techniques described has its limitations and specific deployment, and thus developing an efficient WPT system should not be limited to any single one of the listed techniques [19]. Combining the aforementioned approaches can lead to efficient WPT and a further reduction in the energy consumption of WSNs based on WPT. The current project’s goal is to verify the impact of advanced techniques of WPT on energy transfer performance. This paper is devoted to studying the impact of multi-hop on WPT performance using the designed multi-hop node (MHN). The main focus of previous works on multi-hop wireless power transfer was theoretical and simulation studies to optimize the WPT network. The work [39] determined the efficiency of the energy transfer verses the number of hops, while work [40] presented an optimization model to determine the number of required PBs. The development of MHNs and experimental validation of MHN’s performance in the indoor environment has received much less attention. The work [36] experimentally validated the MHN prototype that generates the power-carrying signal in the form of data packets with a 915 MHz carrier frequency. The system uses off-the-shelf evaluation boards; however, the distance between the PB and the end node is only 30 cm, and only the line-of-sight (LoS) is explored. Another aspect not covered is modeling the performance of the MHN node. Developing models with acceptable accuracy and low resource use is challenging. However, the theoretical models that can estimate the performance of WPT under real-life scenarios have high applicability in the system’s development.

The current paper contributes to many aspects of MHN-based WPT development. First, the work proposes a new concept of MHN based on signal amplification. The presented approach results in a simpler design that can provide greater distance than the previously mentioned signal generation-based approach. The carrier frequency of the WPT system in the study is 865.5 MHz, as this frequency was used in our prior research [41] on RF–DC converter designs for WPT purposes. The emphasis on the sub-GHz range (primarily the ISM bands) is due to the increased distance for power transfer with compatibility with IoT and LORA. For example, commercially-available wireless power transfer-based products by Powercast specifically target these frequencies and applications [42]. It is important to point out that commercially-available WPT products do not utilize the multi-hop approach as of yet. Second, the paper discusses the challenges of MHN design and proposes a method of MHN development. Third, the validation of the MHN is performed using laboratory measurements and simulations emulating real-life WPT scenarios: LoS energy transfer and NLoS energy transfer. A turn-on angle of 90 degrees for MHN is required to mimic a non-direct scenario in a typical indoor environment, like an “L” shaped corridor. Fourth, the paper addresses the challenges of developing simulation models for the provided WPT scenarios and suggests an algorithm based on using different computational methods: the finite element full-wave method for acquiring the far-field antenna patterns, the ray-tracing method for estimating the power transfer efficiency, and the harmonic balance method for estimating the converted DC power.

The paper is organized as follows: Section 2 describes the multi-hop wireless power transfer concept. Section 3 details the developed multi-hop WPT node-design process and parameters. Section 4 presents experimental verification of multi-hop WPT technology, while Section 5 discusses the multi-hop WPT model and proposes an efficient simulation approach for performance evaluation. The final section concludes the work.

## 2. Concept of Multi-Hop Wireless Power Transfer

This section elaborates more on the multi-hop concept in the architecture of WPT and discusses possible implementations of the multi-hop node. The WPT architecture comprises a power transmitter called a power beacon (PB) that transmits the power-carrying signals to the surrounding SNs. This scenario is depicted in Figure 1a. The figure demonstrates that one PB cannot provide sufficient power to all SNs, as some are out of the cover range of the PB. Hence, additional PBs are required. This is not an optimal strategy, as an additional PB may be required to power only a single SN. Additionally, several PBs may interfere in situations where their coverage overlaps. An alternative architecture is illustrated in Figure 1b; it exploits multi-hop nodes to slightly extend the coverage to reach the SNs that otherwise would require an additional PB. This mitigates the interference while simplifying the overall architecture of the WPT network.

The conceptual design of the MHN is defined by how the “hop” is implemented. Figure 2 demonstrates an MHN based on signal amplification. In this method, the RF-DC converted power is stored in a capacitor or a battery to power the connected SN. When the energy storage has sufficient energy for the SN, some is used to power the power amplifier (PA) to amplify the received power-carrying signal and send it further when the PA is active. Then, the received signal is partially amplified and transmitted, with some part of the signal rectified to charge the capacitor. This concept was adopted due to encouraging results reported in one of the previous contributions [43], showing it as a more viable approach for MHN. However, this technique possesses several shortcomings to be addressed. To be more precise, a proper MHN design necessitates balancing multiple requirements, e.g., high output power, low current consumption, high gain, and high RF–DC power conversion efficiency. Additionally, switching between the amplifying and charging modes must be optimized.

## 3. Multi-Hop Node’s Design and Parameters

This section elaborates on the development process of the multi-hop node, outlining the necessary development stages and presenting the final design based on the signal amplification approach. The section also discusses limitations and compromises introduced at each stage of development. The section concludes by demonstrating the fabricated MHN prototype.

### 3.1. Workflow of the Multi-Hop Node’s Development

The MHN based on signal amplification development steps, shown in Figure 3, starts with determining the node’s output power level from the amplifiers. The second step requires analyzing and determining the approximate input power level range either with simulations or experimentally, as these parameters will determine the requirements of the amplifiers and the RF–DC converters used in the design. The third step of development determines the duration of the node’s active (amplification) state. The active node amplifier’s current consumption and storage element’s capacity determines how long the node will be active; the expected input power level will allow for estimating how long the node will charge the storage element to the required threshold for it to operate. After the initial selection of design specifications is finished, the fourth step is devoted to selecting the topology and components and testing the components’ performance. The fifth and final step is the prototyping and validating the complete MHN.

### 3.2. Multi-Hop Node’s Design Limitations and Final Structure

As mentioned previously, the development of the MHN introduced design compromises, as the MHN must satisfy numerous requirements in order to fulfill a useful and efficient solution. The MHN relaying power performance depends on each component’s performance, and the following discussion demonstrates this and how the development steps influence one another.

One of the hurdles introduced in the development is matching the node’s input impedance, as there is an input mismatch between the active state and the state of capacitor charging. The initial design solution was to use solid-state RF switches or RF relays to direct the received signal fully to the rectifier or amplifier circuitry. However, the mismatch when the switch is in an undefined state due to battery storage being in “cold” start conditions introduces additional issues in the matching system for the “cold” startup state and active state. When developing a prototype with discrete components, matching becomes a pronounced problem due to changes in different system operating modes, for example, switching the input signal to the rectifier in the system “cold” start conditions when the system storage voltage level is low or empty. This problem could be partially solved using the power of the storage system; however, it will always require a minimal charge to generate control signals for switches and discharge them. For this reason, the final design does not employ RF switches or RF relays; thus, this creates a compromise between better RF–DC matching and avoiding the uncertainty state which may brick the node. In the current MHN design, the input signal port is split into two: one path to the RF–DC rectifier and the other to the RF amplifier cascade. Input matching is performed specifically for the MHN charging mode, as this state will mostly occur relative to the active MHN state; therefore, with better matching, the more energy the system will harvest. There will be losses due to mismatch when the MHN is in a charging state due to reflections from the PA’s input in the MHN off state. There will also be a mismatch when the MHN becomes active and works in transmitting mode. However, it will have a negligible overall impact on the total harvested amount of energy, as such a state will be relatively short compared to the MHN charging state.

As the amount of the harvested energy depends on the received signal power, it will directly translate to the time required for storing energy. In order to improve the transmission mode duty cycle of the MHN, it is required to improve the RF–DC converter’s efficiency as much as possible and reduce the amplifier’s power consumption while still being capable of further transmitting a considerable amount of power. RF–DC converter matching and tuning were selected for the simulated input power range from modeling; the input power range and the required output power range requirements also narrowed down the chip selection for the amplifier and also introduced a lot of limitations due to requirements for the amplifier: The amplifier must have a considerable gain and produce a high output power at the same time, consuming less current than the counterparts. In this endeavor and comparison of amplifier datasheets, it was concluded that a two-stage amplifier design is required to bridge the necessary gain and output power level with reasonable current consumption to satisfy low power consumption. The PA’s efficiency directly impacts the overall MHN system’s performance, as most of the harvested energy is consumed by the power amplifier stage.

The power amplifiers require a stable supply when in active mode and a higher voltage than the RF–DC converter can provide. For this, an additional DC–DC boost converter chip must be introduced. The DC–DC converter must have a maximum power point tracking (MPPT) function to maximize the harvested power at the optimal operation point. The DC–DC converter will work in the set MPPT operation point, where the maximum power can be extracted from the RF–DC converter. Additionally, the DC–DC converter’s output switching and amplifier turn-on speed must be fast in comparison to the active MHN state duration.

A final MHN structure was formed by performing the MHN development steps and considering the mentioned limitations. As a result, the final MHN in Figure 4 consists of several parts that must fulfill the required functions, such as the RF–DC converter, DC–DC boost converter, a storage element, and a two-stage amplifier consisting of PA_1_ and PA_2_. The RF–DC converter is used to convert the RF signal to the DC signal. The DC–DC boost converter is used to charge the external capacitor from the RF–DC converter output, boost the output voltage, and supply the voltage to the power amplifier cascade through external transistors. The two-stage amplifier cascade is used to amplify the signal from the input of the MHN system and relay it to the ESN. A storage element is required, in this case, a capacitor, to store the harvested energy from the RF–DC converter.

Theoretically, the node can also supply some low-power IoT sensors or WSN nodes; however, it will impact the transmission period of the MHN system, depending on the received power level from PB and the current consumption of the sensor. Such implementation was not reviewed in this paper.

### 3.3. Multi-Hop Node’s Prototype Implementation

After the structure of the MHN was finalized and the requirements for each component were set, the next step was aimed at selecting the components and prototyping the MHN. The schematic in Figure 5 shows the MHN design layout and component selection. The chosen RF–DC converter is based on the well-known and extensively used voltage doubler topology with two Schottky diodes SMS7630 [41]. These diodes are well-suited for high-frequency applications, owing to the low voltage drop they exhibit. The RF–DC converter output voltage is boosted using a DC–DC switching converter BQ25570 and stored in a 1 mF capacitor connected externally. As for the amplifiers, the first stage is the Qorvo RF2878 amplifier, while the second stage amplifier is Guerrilla RF GFR5020.

The fabricated MHN prototype is shown in Figure 6. The prototype is a compact layout on a 1.6 mm FR4 base, with the board itself being 5 by 5 cm. The node’s prototype was constructed so that most of the individual parts’ performance could be measured, which is elaborated more in our previous work [43].

## 4. Experimental Verification of the Multi-Hop Wireless Power Transfer Technology

To prove the validity and demonstrate the benefits of the proposed concept, two indoor wireless power transfer scenarios are examined: LoS energy transfer and NLoS energy transfer. In the LoS scenario outlined in Figure 7, the PB’s, MHN’s, and ESN’s antennas are all lined up. This experimental study compares the received (rectified) power level on the ESN with the MHN to the case without the MHN (only PB and ESN).

All antennas employed are four-element Yagi–Uda designed to operate at 865.5 MHz. The antenna gain at this frequency is 9.18 dBi. The Yagi–Uda antennas are employed due to high gain and narrow beam, reducing the multipath propagation’s impact. The distance between the antennas is measured from the first dipole, which is approximately 10 cm from the edge of the antenna. The distance between the PB’s transmitting antenna and the MHN’s receiving antenna is set to 3.25 m. The distance between the MHN’s transmitting antenna and the ESN’s antenna changes from 1.15 m to 3.4 m. All antennas are located at the same height of 1.07 m.

In the NLoS scenario outlined in Figure 8, the imaginary straight line passing through the PB’s antenna and the receiving antenna of the MHN subtends a right angle with that passing through the transmitting antenna of the MHN node and ESN’s antenna. The antennas and the distances are similar to the LoS scenario. The key difference in this study is that the turn angle between the PB and ESN is 90 degrees; thus, there is no direct line of sight between these nodes. Due to the NLoS and the narrow beam of the antennas, this experimental study evaluates the performance only in the case of the MHN.

The next subsection elaborates more on the ESN’s RF–DC converter prototype used to evaluate the impact of MHN on the WPT performance, followed by the measurements setup used for both WPT scenarios and the analysis of the acquired results.

### 4.1. End Sensor Node’s RF–DC Converter Prototype

The ESN prototype’s RF–DC converter used in this study is the voltage doubler-based design shown in Figure 9. The design is based on the Skyworks SMS7630 diodes. The fabricated prototype in Figure 10 is a compact solution fabricated on a 1.6 mm FR4 base. This design was explored in our works [41,44] and we proved its efficiency. The latest iteration of this design uses low-DC-resistance inductors from Coilcraft, allowing for reaching significantly higher power conversion efficiency (PCE).

The PCE of the RF–DC converter is also highly dependent on the load resistance. Figure 11 demonstrates the dependence of the PCE on the load resistance of this converter at various input power levels. The following formula is used to calculate the PCE:(1)$$\mathrm{PCE}=\frac{P_{rec}}{P_{in}}\times 100\%,$$
where *P_rec_* is the rectified power on the output of the RF–CD converter, and *P_in_* is the input power of the RF–CD converter.

At the input power level of −20 dBm, the power conversion efficiency reaches 23.7% with 5 kΩ load resistance, efficiency at −15 dBm input power at 5 kΩ load resistance reaches 38.4%, efficiency at −10 dBm input power at 4.3 kΩ load resistance reaches 52.5%, efficiency at −5 dBm input power at 4.2 kΩ load resistance reaches 63.8%, efficiency at 0 dBm input power at 4.4 kΩ load resistance reaches 70.9%, and efficiency at 5 dBm input power at 3.3 kΩ load resistance reaches 75.7%. For the use cases with the MHN, the rectifier will be used in the power range from 0 to 5 dBm; therefore, the optimal load resistance is 3 kΩ. After defining this crucial parameter of the ESN’s RF–DC converter, the ESN can be used in the experimental study.

### 4.2. Experimental Measurements Setup

The experimental study on the LoS propagation scenario is performed with the setups depicted in Figure 12 and Figure 13. Figure 13 demonstrates the setup used to measure the received power level and rectified voltage at the ESN and its dependence on the distance between the MHN and ESN. As the setup shows, the 865.5 MHz power-carrying signal on the PB is generated by the R&S SMC100A generator through the power amplifier (PA) MMG3006N by NXP, resulting in the 26 dBm output power. The signal waveform from the ESN with active MHN is saved with the Tektronix DPO72004 oscilloscope (Salt Lake City, UT, USA) and later processed to calculate the average power absorbed into the oscilloscope’s 50 Ω input termination, considering the SMA cable attenuation. The voltage across the RF–DC converter with 3 kΩ load resistance at the end node is measured with Digilent Analog Discovery 2 (AD2). In this case, the cable attenuation to the oscilloscope is also considered in the calculations. The setup in Figure 13 is similar but without the MHN, and the distance is measured between the PB and ESN antennas.

For the NLoS scenario, only the experimental setup demonstrated in Figure 13 is used for measuring the received power and rectified voltage at the ESN, as the NLoS scenario requires the MHN for operation.

### 4.3. Experimental Results Analysis

The results of the experimental study are compiled in Figure 14 and Figure 15. Figure 14 demonstrates the dependence of the ESN’s received power level and rectified voltage on the distance between the PB and ESN with and without MHN. The case of LoS with MHN shows a steady power level decrease on the ESN from 11.66 dBm to 1.67 dBm in the distance range from 4.9 m to 6.25 m. However, from the distance of 6.25 m to 7 m, the received power level shows a reverse trend, as the power level increases up to 3.42 dBm at the 7 m distance and a sharp decline at the 7.15 m with the received power level of 2 dBm. The rectified voltage level follows the same trend as the received power level: a steady decrease from 5 V to 1.74 V across the 3 kΩ load resistance in the distance range from 4.9 m to 6.25 m, then a slight increase in the rectified voltage level from 6.25 m onwards, reaching the 2 V voltage level at 7 m and decreasing at 7.15 m with a 1.6 V level. Thus, the use of MHN, even in the presence of the line-of-sight propagation path, has been shown to sufficiently increase the amount of received power. The use of MHN in LoS conditions in the active state increases the received power level from 5.5 dB to 19.16 dB depending on the position from the PB, where the lowest difference in power levels is measured at 6.25 m and the largest at a distance of 6.85 m. The difference in the rectified voltage at the ESN with MHN in active state is significant; the voltage increase relative to the rectified voltage without MHN is from 203% to 3189%. MHN allows the receiving of significantly higher power levels and, therefore, higher rectified voltage compared to non-MHN measurement scenarios.

Figure 15 presents the measurement results for the NLoS scenario. The received power level follows the same trend as in the previous scenario, where the received power level decreases linearly from 11.21 dBm to 0.92 dBm in the distance range from 4.9 m to 6.55 m. From 6.55 m to 7 m, the received power level fluctuates around the same value and, in the end, drops to a −0.34 dBm power level. The rectified voltage level follows the same trend as the received power level: the voltage level linearly decreases from 4.84 V to 1.52 V in the distance range from 4.9 m to 6.55 m, and from 6.55 m to 7 m fluctuates in the same voltage range and drops at the last measurement point to 1.32 V.

The ratio of the ESN harvested energy and MHN transmitted power also depends on distance. In the LoS case, the ratio is 3.7% at 4.9 m and 0.4% at 7.15 m. In the NLoS case, the ratio is 3.3% at 4.9 m and 0.2% at 7.15 m.

The acquired results not only show the benefits of using MHN in the two different WPT scenarios but also demonstrate that the received power is highly dependent on the ESN’s position. When developing a WPT network, figuring out the optimal positions of the nodes via experimental measurements is costly. This highlights the need to develop WPT models that sufficiently match real-life measurements while demanding the least computational time and resources. This is the next step of this research and is described in the next section.

## 5. Model Development for Multi-Hop Wireless Power Transfer

Fast and accurate PB-to-ESN power transfer estimation is of primary concern in WPT systems antenna positioning optimization seeking such configurations of a WPT system deployed in some indoor environment that minimizes the amount of wasted power. The numerical modeling of power transfer channels obviates the need for experiments that, in some cases, may require expensive instrumentation.

While in sparse outdoor environments it is possible to estimate power transfer efficiency based on the well-known Friis transmission equation [45] or some refinements and extensions thereof [46,47,48], in indoor environments, even those with relatively simple geometry, the power they use of it may lead to dramatic discrepancies between the theoretical results and those obtained through measurements. The main source of the discrepancies is the fact that this equation does not consider multiple reflections off the walls, as well as other elements of the indoor environment under study. Furthermore, the highly approximate theory based on which the Friis equations are derived completely ignores various diffraction mechanisms that may contribute considerably to the discrepancy. This is particularly the case in environments featuring a number of objects with sharp edges [49].

The presence of walls, ceiling, floor, and windows, as well as plenty of different cabinets and other office elements, make the estimation process quite a challenging problem that is not easy to address using the existing methods, especially when the wavelength at a working frequency is not ten or more times smaller than the environment (e.g., an office room) dimensions but is comparable with those of objects located in it and furniture and decoration elements, such as a suspended ceiling. Multipath propagation resulting from multiple reflections of the transmitting (TX) antenna radiated waves from the walls and various objects present in the environment result in interference that, in turn, leads to variations in the received amount of power with the antenna position. In some cases, the variations may be so significant that a slight receiving (RX) or TX antenna shift might affect the received power.

In such a case, the modeling accuracy can be improved by applying some empirical or theoretical corrections to the Friis model. Although introducing corrections to the Friis model may seem simple to implement at first glance, a closer examination of this approach reveals that it may not always be the most optimal way. A more accurate approach to modeling indoor environments is the one based on the use of numerical techniques exploiting high-frequency approximations of wave equations, such as the geometrical optics approximation (GO, also commonly known as the ray-tracing method) [50,51,52] and the physical optics approximation (PO) [53,54]. However, this approach may fail to accurately predict the behavior of real-life models because of their approximate nature, albeit being very fast.

In an attempt to improve accuracy while maintaining an acceptable computational burden when analyzing large-scale real-life problems, various extensions and combinations of both PO and GO, as well as other methods, have been developed over the last few decades [55], e.g., the shooting and bouncing rays method (SBR) [56,57] and the hybrid PO–SBR–PTD Method [58]. Both plain GO (ray tracing) and PO methods have been successfully applied to solving a wide variety of practically important problems, such as radio wave propagation channel parameter estimation in both outdoor and indoor environments, e.g., tunnels [59,60,61].

However, in the case of environments involving objects with sharp edges, even refined high-frequency methods may still fail to provide acceptable accuracy when the diffraction effect is pronounced. To mitigate this issue, a few formulations that approximately describe the underlying diffraction mechanisms have been developed [62]. The geometrical theory of diffraction, originally developed by Keller [63] and subsequently extended by several researchers, mitigates the inability of the geometrical optics approximation to describe the fields in the region of geometrical optics’ shadow, which fails to describe the fields resulting from the diffraction effect at edges, tips, vertices, and other object surface discontinuities. Furthermore, the original Keller’s theory, as well as its extensions, are capable of ensuring sufficient field approximation accuracy relatively far from the objects the incident field interacts with due to the fact that they are derived from approximate closed-form expressions under the high-frequency assumption. The method is based on approximating the diffracted fields in terms of a set of rays consisting of two straight lines meeting at points of a sharp edge the incident rays hit or other geometrical discontinuities. However, it was shown that the GTD fails to ensure uniformity in the transition regions separating the reflection shadow and diffraction shadow regions. Later, in an attempt to overcome this issue, an improved formulation was proposed by Kouyoumjian and Pathak [62], which is nowadays commonly known as the uniform theory of diffraction. Similar to Keller’s GTD, the UTD also approximates diffracted fields as a bunch of rays obeying an extended version of the Fermat principle. While the UTD can provide a more consistent approximation of diffracted fields, the ray field coefficients (diffraction coefficients) in this case are also derived under the assumption that the observation point is many wavelengths away from the object under consideration. At smaller distances, both the GTD and UTD may provide highly inaccurate results. Therefore, one cannot expect that the two approximate high-frequency techniques will ensure acceptable field computation accuracy when a model being studied involves many relatively closely-spaced fine elements with edges.

An alternative diffraction field approximation technique applies a correction to the physical optics approximation results to improve the field calculation accuracy. The technique is termed the physical theory of diffraction (PTD) in the scientific literature [64,65], and the correction to the surface currents used to compute the scattered field is derived using the same reasoning as the one used to derive the GTD diffraction coefficients. As the correction is deduced based on the same reasoning as in the case of the GTD and the UTD, the PTD has the same shortcoming; namely, it is not guaranteed to provide an adequate diffracted field approximation when a diffraction-causing object (e.g., suspended ceiling fixture) and other objects, such as the ceiling in the present study are not well separated. To summarize, in the case of indoor environments with complicated structures, the above-mentioned methods may fail to provide adequate results due to their approximate nature. In such a situation, the last resort is to employ general-purpose full-wave analysis methods, e.g., the finite element method (FEM) [66,67] and the method of moments (MoM) [68]. The latter is also known as the integral equation (IE) method. Despite their remarkable flexibility and capability to handle structures with arbitrary geometry no matter how complicated it may be, they are, unfortunately, tremendously computationally expensive, and the CPU time grows very quickly as the dimensions of the environment model increase, for example, when applied to models whose dimensions are multiples of a wavelength [69,70].

It is worth noting that, in some cases, the computational burden can be reduced by the use of semi-analytical methods. These methods can be applied when the environment can be decomposed into several regions, some of which have simple geometry that enables one to exploit field description in terms of the entire domain basis function well-suited to the given geometry, such as cylindrical functions, spherical, etc. [71].

Considering the benefits and pitfalls of high-frequency and full-wave techniques, in the present paper, the antenna-to-antenna power transfer is analyzed using two approaches, and the obtained results are compared with each other, as well as against the measurement data acquired in a real-life indoor environment (an office room). This comparative study’s main objective is to determine how close the numerical results are to the experimental ones for a given indoor WPT system model.

The ray-tracing method-based model of the indoor environment is examined first, then the received power level for an indoor environment, the laboratory room where the measurements of the MHN-assisted WPT systems take place. Then, the same model is analyzed using Ansys HFSS, which solves Maxwell’s equations employing general-purpose numerical methods, including the FEM or MoM.

While the ray-tracing method is approximate, the wavelength at the operating frequency is approximately 34.6 cm, which is much smaller than the room dimensions, which implies that the simulation results may provide a reasonably good approximation of the actual field distribution while requiring much less CPU resources compared to the full-wave analysis based power level estimation, as the full-wave simulation for such an electrically (relative to the wavelength) large model would lead to an amount of CPU time prohibitive for off-the-shelf computers.

### 5.1. Ray-Tracing Method-Based Analysis

As far as the ray-tracing method is concerned, in the numerical studies described herein, not only the line-of-sight path but also reflections from the room side walls, ceiling, and floor are taken into account, including multiple reflections; a maximum of two reflections from the model sides are considered. Namely, only the waves that have undergone no more than two reflections from the room sides before arriving at the receiving antenna are assumed to contribute to the total received wave, which is nothing but a superposition of different multipath components.

The ray-tracing-based modeling procedure comprises two stages. In the first stage, the far-field pattern for a four-element Yagi–Uda antenna is computed. Additionally, the ray-tracing method requires far-field data for the antenna-to-antenna power transfer calculation. The corresponding HFSS model is constructed for the antenna, and the far-field pattern is calculated using the finite element full-wave solver incorporated into the software.

By performing multiple antenna simulations for different parameter value sets, the optimal variation from the maximum gain point of view has been found, and four identical antenna prototypes were fabricated. These antenna prototypes were utilized to validate the multi-hop node-assisted power transfer systems. Both the E-plane and H-plane radiation patterns for the gain, which factors in the antenna input reflection, were computed and are illustrated in Figure 16.

Once the far-field patterns are found, they are exported from HFSS into MATLAB for the indoor environment under study ray-tracing. In the second stage, the transferred power from the transmitting antennas to the receiving one is calculated using the GO approximation. Specifically, the field radiated by a transmitting antenna is treated as a set of rays emerging from a point source representing the antenna. The parameters of the ray-tracing model are summarized in Table 1. In this model, the floor and ceiling are assumed to be infinite in extent; in this case, such an assumption does not significantly affect the results.

The receiving antenna is also assumed to be a point receiver. It is worth noting that the aperture area is calculated as the product of the isotropic antenna aperture and the antenna gain in a specific direction, e.g., the direction of arrival of an incident wave (ray). For each multipath component, the path is determined based on the location of the phase centers of both the receiving and transmitting antenna and reflection points that can be readily found using simple geometrical methods. Each time a ray bounces off the model surface, its phasor is multiplied by the corresponding reflection coefficient calculated based on the Fresnel transmission equations.

The received power levels against the separation between the antennas computed using Ansys HFSS (full-wave) and the ray-tracing method are presented in Figure 17. The curve obtained for free space is plotted on the same graph as the ones obtained using Ansys HFSS and the ray-tracing method. It is worth noting that the effect of the suspended ceiling is not accounted for by the ray-tracing method, as outlined above. Both the full-wave and the ray-tracing models treat the ceiling and the floor as conducting planes with sufficiently high conductivity. In this study, the conductivity of iron is assumed.

It should be noted that, in the theoretical analysis, the modeling of the amplifying node is not carried out. The effect of this node on the rest of the system is incorporated through the node gain and the total phase shift between the sine waves at the node’s input and output. While the importance of the former node parameter is obvious, the role of the latter one may seem not essential at first; however, it is not the case. Specifically, the phase determines the shape of the received power curve in the case of the LoS scenario, as the total power received by the ESN is not determined by the power radiated by the MHN node and the separation between the nodes. Still, it is the power of the superposition of two waves, one being the wave due to the MHN node, whereas the other is the wave coming from the PB.

The power level of this direct component is smaller by an order of magnitude than the one received from the MHN. However, the phase difference between the waves depends on the distance between them because the direct wave is composed of several waves: the LoS one and the waves reflected from the ceiling, floor, and walls. Due to this difference, the shape of the curve varies with the phase delay introduced by the node; as a result, the curves obtained under LoS and NLoS scenarios may differ somewhat.

Another set of curves is shown in Figure 18. In this case, the full-wave model is the same as in Figure 17, but the ray-tracing model is slightly modified. More precisely, the effect of the suspended ceiling is approximately accounted for via the introduction of the effective antenna axis to ceiling distance, approximately 11.5 cm shorter than in the case without a ceiling. The main idea of this approach is based on the fact that the metallic grid of the suspended ceiling exhibits a non-zero transmission coefficient at incident angles that are not sufficiently large. Consequently, some fraction of a wave impinging on the grid passes through it, and then it undergoes multiple bounces between the actual ceiling surface and the grid, which results in multiple reflected waves with different amplitudes and phases that form the total reflected wave. The amplitude of the total reflected wave is still close to unity if the ceiling is assumed to be conducting, but the phase may differ considerably from the one reflected from the ceiling surface only. Moreover, its dependence on the angle of incidence differs from that governed by the classical Fresnel equations. It is possible to achieve the required phase for a specific angle by changing the distance between the ceiling surface and the antenna axis. However, the change should not be large to not affect the wave amplitude due to the variation in the wave travel distance.

As evidenced in Figure 18, the effective distance approach provides better results; the large minima in both curves coincide in this case. However, the curve shape differs considerably between the minimum and the maximum, occurring at approximately 3.5 m and 4.2 m, respectively. The discrepancy is likely due to the dependence of the effective distance on the incidence angle.

In general, finding an effective antenna axis to ceiling distance in many cases may not be a trivial task, and it is not guaranteed that the approach based on the effective distance concept will ensure acceptable power level estimation accuracy; moreover, it is beyond the scope of the present investigation, and a more in-depth treatment of this approach will likely be provided in one of our forthcoming papers.

### 5.2. Full-Wave Analysis

To perform a full-wave analysis of the indoor environment under study, Ansys HFSS is employed. The main advantage of the full-wave analysis is that it takes into account the effect of the suspended ceiling metallic elements (metallic grid) on the wave propagation directly without using any physical or geometrical optics or other approximations. Ignoring this effect might lead to a dramatic discrepancy between the estimation and measured results, as was shown in the previous subsection on the ray-tracing analysis. However, direct full-wave analysis is highly computationally intensive. Therefore, the indoor environment model examined in this study is significantly simplified to reduce the number of model discretization elements, thus reducing CPU time. To be more precise, only the ceiling and floor, including the fixtures of the suspended ceiling (metallic grid), are accounted for in the model, while the effect of the wall is completely ignored. This simplification is justified by a simple investigation of the radiation pattern of the narrow beam antenna used in the study and the behavior of the waves impinging on the floor, ceiling, and side walls, which is governed by the Fresnel equations. According to the Fresnel equations for the parallel incident plane wave polarization (TE case), the waves impinging on the side walls with a small grazing angle (due to the narrow antenna beam) have a small reflection coefficient. Thus, the contribution of the wave reflected from the side walls can be ignored, and therefore, they can be removed from the model. The flowchart of the entire procedure for full-wave analysis of the indoor environment is depicted in Figure 19.

To reduce the computational burden, the indoor environment model is divided into three parts: the receiving antenna model, the transmitting antenna model, and a model of the environment. Each of the three sub-models of the original model is treated separately. For the full-wave analysis of both receiving (RX) and transmitting (TX) antenna models, the finite element method solver is employed. The solution space is truncated using absorbing boundary conditions (ABC), which in the HFSS are available as the radiation conditions option. The conditions imitate the absorption of the antenna-radiated waves. These boundary conditions are less computationally expensive than their PML and Hybrid FEM–IE counterparts. However, it comes at the expense of higher reflection from the absorbing boundary when the absorbing surface is not sufficiently far from the antenna model itself. The dimensions of the box to the surface of which the above-mentioned boundary conditions are assigned are as follows: width—390 mm, length—890 mm, and height—200 mm. These dimensions were determined empirically by performing a series of numerical studies with gradually increasing the bounding box dimensions. Additionally, a fictitious surface enclosing the antenna model is created. No boundary conditions are assigned to the surface; it is intended to be utilized as an equivalent radiation surface in the second part of the model, which is intended to be linked with this one. First, the fields are calculated for the TX model, including the ones on the fictitious inner surface. Then, the fictitious surface, together with the field distribution on it computed by performing simulations of the TX antenna model, is transferred to the next model, namely, the room model, where it will be used as an excitation. The created antenna model of the Yagi–Uda antenna is depicted in Figure 20.

The Ansys HFSS model of the indoor environment (room) is shown in Figure 21. In the room model, a dummy region of space where the TX antenna model would otherwise be located (TX antenna placeholder) is created and treated as being filled with the perfect electric conductor (PEC) to prevent the solver from seeking the fields within it.

The floor and the ceiling are modeled as flat rectangular surfaces, which are assigned impedance boundary conditions and are defined as hybrid IE regions. Specifically, the ceiling and floor models are handled by an integral equation solver (IE solver) that employs the MoM for surfaces. The surface impedance value corresponds to that of iron. The model of the metal grid (suspended ceiling) is a solid 3D object on the surface of which other impedance boundary conditions are imposed. This object is also treated by the IE solver. As mentioned above, this room model does not contain the model of the TX antenna; in place of it, another fictitious surface is created whose dimensions are slightly larger than those of the fictitious surface used in the RX antenna model. The region of space inside the surface may be regarded as a placeholder for the TX antenna. The surface is necessary for computing fields on it.

Once the fields on the second fictitious surface are found, they are transferred through the link established between the room model and that of the RX antenna and used to illuminate the TX antenna model as if it were located in the room model. The RX model is identical to that of the TX antenna. The parameters of the full-wave model under study are presented in Table 2.

It is worth noting that the software exploits the surface equivalence principle stating that the antenna-generated fields outside the surface can be evaluated knowing the field on some surface enclosing it. This eliminates the need to discretize the room model part corresponding to the TX antenna. However, handling the indoor environment model in this way ignores the two-way interaction between the antenna and the rest of the model. Nevertheless, since, in the present case, the interaction is not strong, mainly affecting the waves reflected back to the antenna, which is not of concern in the present study, the partitioning of the original model will only slightly influence the final results.

The suspended ceiling used in most modern office facilities is a periodic metallic structure intended to hold stone wool slabs whose thickness typically ranges between 2 and 5 cm. The dielectric constant of the slab material in the indoor environment under study is approximately 1.2–1.5, and it has a negligible effect on the wave propagation, which was experimentally verified in the present study; placing a single slab between a pair of Yagi–Uda antennas does not affect the received power level under normal incidence. However, when two such stone wool slabs are stacked together, the amount of the received power is reduced by about 0.1 dB, which is still very small.

The calculated gain of the antenna is approximately 9.48 dBi, while the measured gain of the antenna prototype is in the vicinity of 9.17 dBi. Such a mismatch between the results might arise from the imperfections of the milling machine used to fabricate the prototype. Among other factors that might cause this may be the difference between the actual value of the substrate material dielectric constant and loss tangent from the ones assumed for the antenna design, some parasitic effects of the SMA connectors used, small inhomogeneous in the substrate material, etc. For the calculated results to be closer to the measured ones, the calculated antenna gain is rescaled so that the antenna model’s maximum gain (in the boresight direction) matches the measured one.

Figure 22 displays the results obtained using the full-wave analysis and the experimentally measured ones in the case of direct antenna-to-antenna power transfer. The figure clearly shows that the results are in close agreement, which means that the full-wave analysis-based estimation, even with a simplified model such as the one used in the present study, may provide reliable power level estimation, enabling antenna placement optimization to increase the overall WPT systems efficiency.

The results calculated for the case of the direct power transfer and the MHN-assisted power transfer under the LoS scenario where all the antennas are lined up are indicated in Figure 23. The power in the LoS scenario is calculated by adding phasors of both the direct component from the PB and the one amplified and subsequently radiated by the MHN. The phasors are computed using Ansys HFSS, while the phase delay due to the node circuitry and the cable connecting the node’s antenna is established experimentally.

The full-wave simulation results obtained for the multi-hop assisted wireless power transfer in the case when there is no line-of-sight channel between the PB and ESN antenna are displayed in Figure 24. Similar to the previous case, two Yagi–Uda antenna pairs are used. The antennas are arranged so that the direct component due to the coupling between the PB antenna and the ESN antenna is eliminated. Alternatively, the two imaginary straight lines, one of which passes through the phase centers of the PB antenna and the receiving antenna of the MHN and the other one passes through the those of the MHN transmitting antenna and the one of the ESN, make a 90-degree angle. The received input power is calculated by combining the full-wave simulation results obtained using Ansys HFSS.

While it would be more favorable from the modeling accuracy perspective to perform analysis of an indoor environment model incorporating all four antennas, the equation system resulting from the model discretization would be huge, making the CPU time unacceptably large. Instead, in this study, the power transferred from one antenna to another is calculated for a two-antenna model, like in the case of the direct power transfer. Then, using the obtained data, the power received by the receiving antenna of the MHN is calculated, multiplied by the node amplifier gain, and eventually, the power transferred to the end node is found using the calculated amount of power transmitted by the MHN and the relation between the transmitted and the received power for the pair of antennas. The last step of the calculation procedure is identical to the first one, as the same data are used (calculated using HFSS). The effect of the antenna position relative to the suspended ceiling grid elements on the relation between the RX and TX power levels is found to be sufficiently little to neglect, which makes it possible to use the data acquired for only one antenna position.

### 5.3. Calculated and Measured Result Comparison

The numerical (modeling) results and the experimentally obtained ones discussed in the previous sections are presented on the same graphs for comparison. The results obtained under the LoS scenario are shown in Figure 25, while those corresponding to the NLoS scenario are observed in Figure 26.

As can be seen, the discrepancy between the calculated and the measured results shown in Figure 25 and Figure 26 is small, even though a simplified model was employed for the full-wave analysis of the indoor environment under study. The input power level of the transfer system under study was kept fixed at the level of 26.0 dBm throughout the investigation, regardless of the power transfer scenario. In the case of direct power transfer from the PB to the ESN, a distinct dip is observed at a distance of approximately 6.8 m. This received power blockage occurs due to the wave reflection off the floor. Since the beam of the antenna utilized in the present study is sufficiently narrow (about 60 degrees), the fluctuation of the received power level about the theoretical curve for free space calculated according to the well-known Friis transmission equations is not significant up to about 6.0 m where the basin of attraction of the minimum begins. It should be noted that if antennas having a much wider beam in the H-plane, e.g., an LDPA, dipole, or monopole antenna, were used, the minima and maxima would be appreciably more pronounced, thereby severely limiting the use of such antennas in indoor power transfer systems due to the antenna positioning complexity. Although this issue could be mitigated by exploiting the spatial diversity of MIMO or similar beamforming approaches, it would be much more expensive than deploying antenna systems based on the printed Yagi–Uda antenna equipped with a small autonomous power amplifier, as treated in the previous sections. The other two extrema, a minimum and a maximum occurring at 3 and 4 m, respectively, are much smaller than the large one at 6.8 m and should not considerably affect the power transfer system performance.

Regarding the difference between the theoretical and practical power level curves, Figure 25 shows that the largest deviation of the measured power level from the calculated one occurs in the vicinity of a maximum in the experimental curve at approximately 5.5 m and the adjacent maximum at 6.4 m. Interestingly, the theoretical curve exhibits no maxima over the antenna separation range from 4.8 to 6.4 m. The absence of the maxima may be attributed to the effect of some fine model elements that were not taken into account to simplify the model, thus reducing the amount of CPU time as described above. For instance, the presence of the lighting elements, including wires, fastening components and fixtures, was not accounted for by the model in the present study.

## 6. Conclusions

This research was devoted to the performance evaluation of multi-hop (MH) technology for wireless power transfer (WPT) at sub-GHz frequencies. The study presents an approach based on using an additional node for path loss compensation, a multi-hop node (MHN). The proposed approach was experimentally validated under real-life scenarios for line-of-sight and non-line-of-sight propagation conditions. However, to apply this prototype for powering IoT, LORA devices will need further optimization, like fine-tuning the time of active and charging modes to meet the power consumption of the powered device. Additionally, it was demonstrated that it is still possible to accurately estimate the power transfer efficiency in an indoor environment with a metallic grid (suspended ceiling) in a reasonable timeframe when the classical ray-tracing method fails to provide adequate modeling accuracy. The numerical analysis of the PB to ESN power transfer under different indoor propagation scenarios was carried out using a simplified full-wave model. The fields were computed using commercially available software Ansys HFSS. In addition, the results obtained using HFSS were compared with those obtained by means of the ray-tracing method. The far-field patterns required for the ray-tracing analysis were obtained using HFSS. It was found that even in the case of a simplified full-wave analysis model, the HFSS results are much closer to the experimental ones than those provided by the ray-tracing method.

Simulation and experimental study results of MHN-assisted WPT have shown that MH technology can sufficiently increase WPT efficiency. In the line-of-sight case, at a distance up to 7 m, the received power level at the end node was increased up to 19 dBm compared to the direct WPT without the MH. Furthermore, the results obtained under the non-line-of-sight scenario show that the MH approach enables sufficiently efficient power transfer even when the direct link between transmitters and the receiver antennas is absent. Indeed, the proposed MH WPT simulation approach could be applied to optimizing the spatial distribution of MH WPT system elements in real-life scenarios.

## Figures and Tables

**Figure 1 sensors-23-07367-f001:**
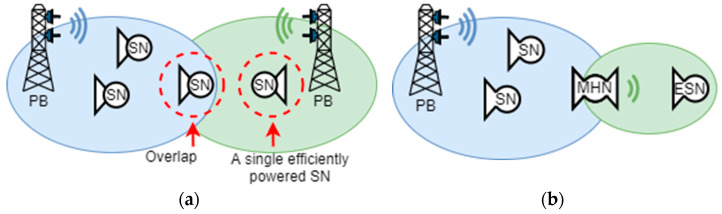
WPT architecture based on individual power beacons (**a**) and WPT architecture based on multi-hop power transfer (**b**).

**Figure 2 sensors-23-07367-f002:**
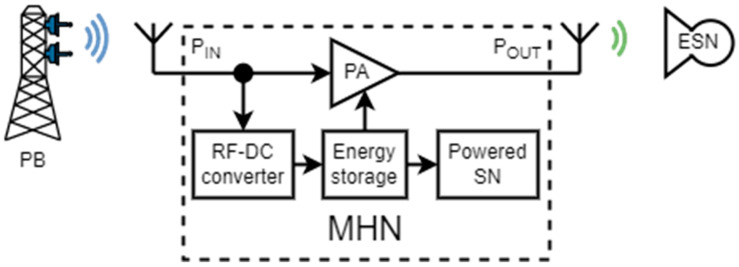
Multi-hop node based on signal amplification.

**Figure 3 sensors-23-07367-f003:**
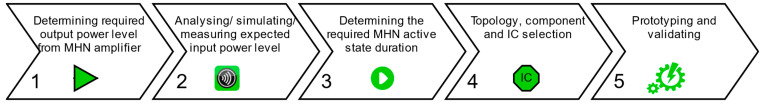
The MHN development process’s block diagram.

**Figure 4 sensors-23-07367-f004:**
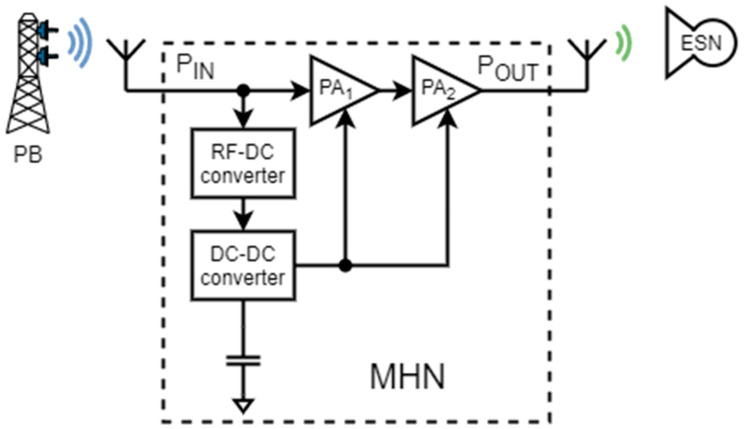
The structure of the multi-hop node.

**Figure 5 sensors-23-07367-f005:**
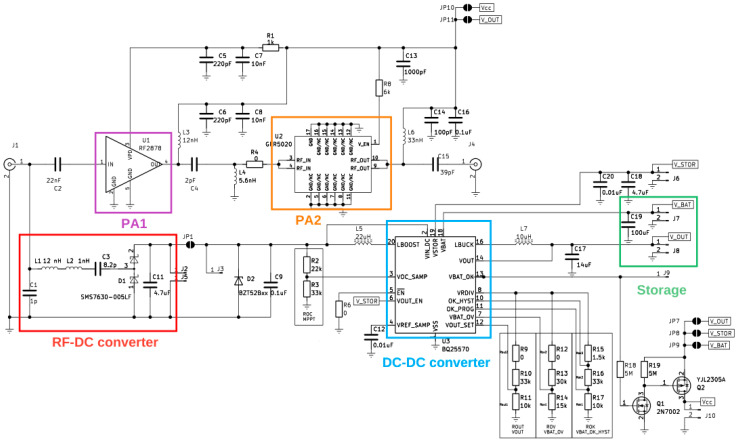
Schematic of the multi-hop node prototype.

**Figure 6 sensors-23-07367-f006:**
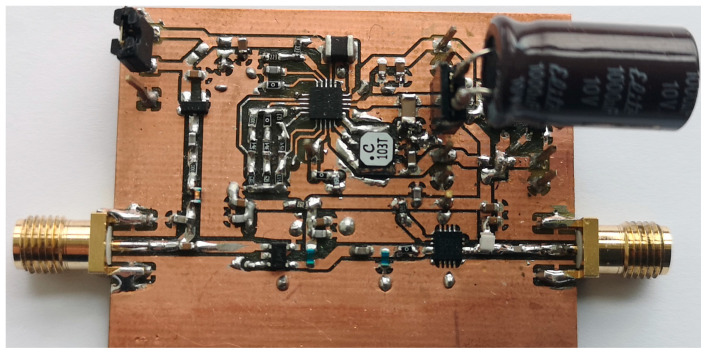
Photo of the multi-hop node prototype.

**Figure 7 sensors-23-07367-f007:**
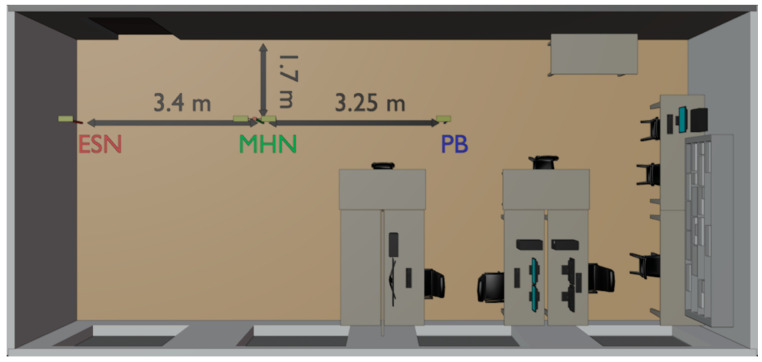
Layout of the indoor environment for studying the LoS MHN-aided WPT.

**Figure 8 sensors-23-07367-f008:**
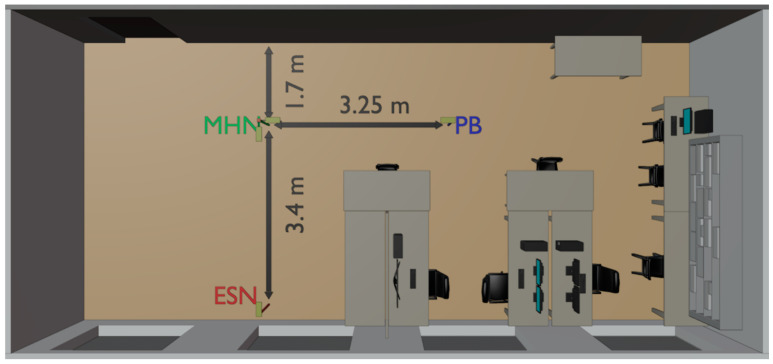
Layout of the indoor environment for studying the NLoS MHN-aided WPT.

**Figure 9 sensors-23-07367-f009:**
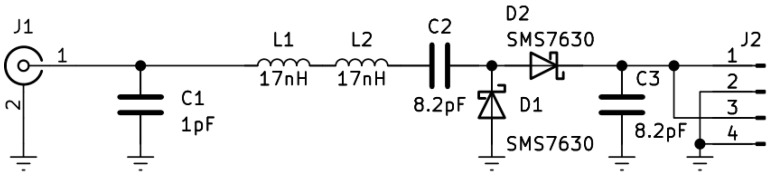
Schematic of the end node’s RF–DC converter.

**Figure 10 sensors-23-07367-f010:**
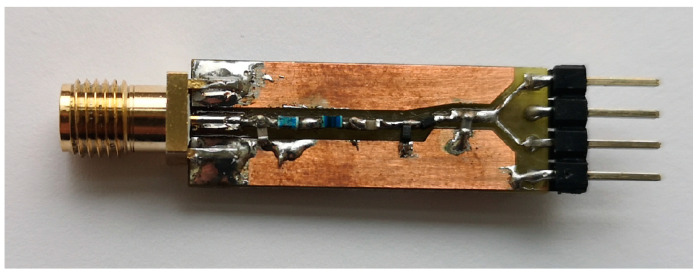
Photo of the end node’s RF–DC converter.

**Figure 11 sensors-23-07367-f011:**
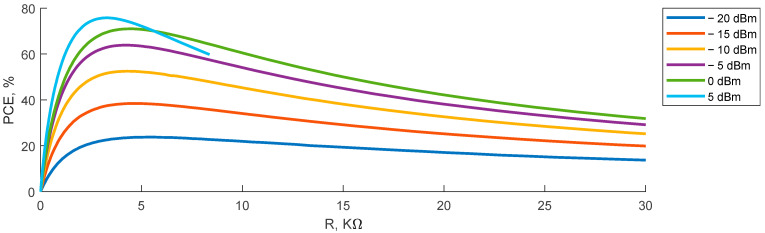
The PCE of the voltage doubler rectifier (end sensor node rectifier) depending on load resistance at different input power levels.

**Figure 12 sensors-23-07367-f012:**
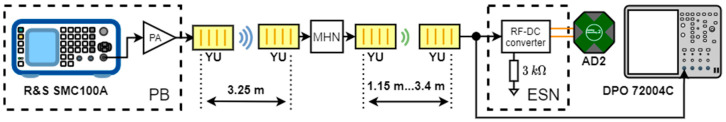
Experimental setup for measuring the received power and rectified voltage at the ESN depending on the distance between the MHN’s and the edge sensor node’s antennas.

**Figure 13 sensors-23-07367-f013:**
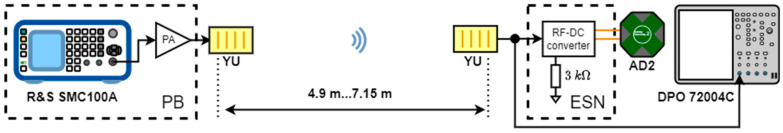
Experimental setup for measuring the received signal power and rectified voltage at the ESN from the transmitter depending on the distance between the antennas (without MHN).

**Figure 14 sensors-23-07367-f014:**
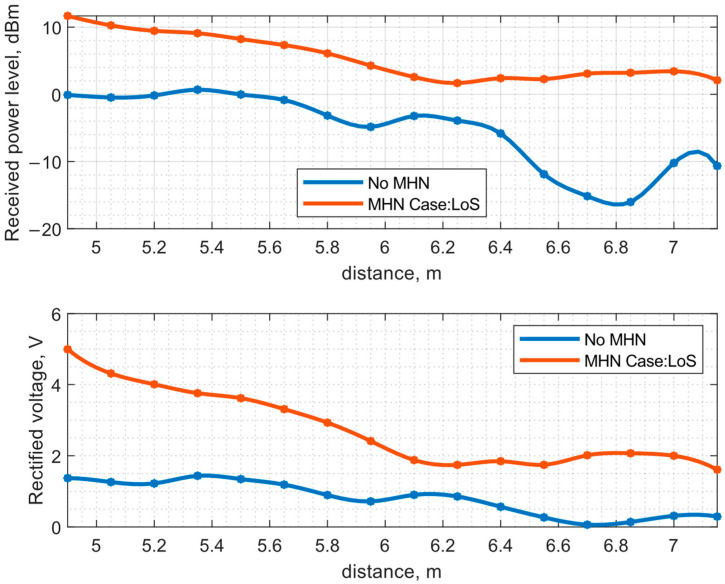
Measurement of received power level and rectified voltage with voltage doubler at the ESN antenna depending on distance between the PB position and ESN in LoS scenario.

**Figure 15 sensors-23-07367-f015:**
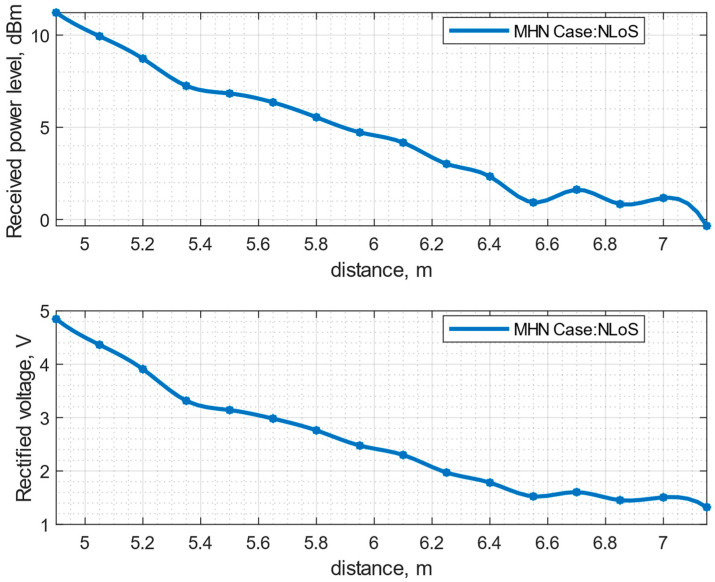
Measurement of received power level and rectified voltage with voltage doubler at the ESN antenna depending on distance between the PB position and ESN in NLoS scenario with MHN.

**Figure 16 sensors-23-07367-f016:**
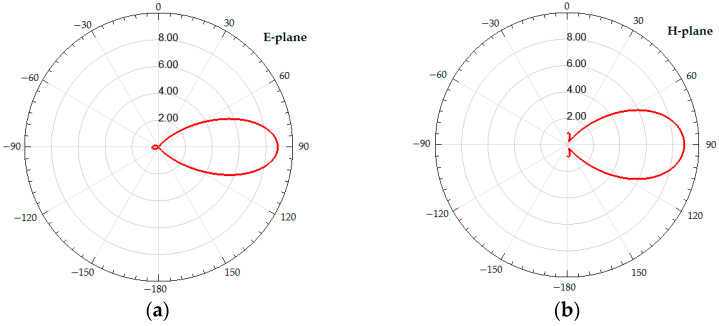
The calculated radiation pattern (calculated using Ansys HFSS) of the Yagi–Uda antenna in the E-plane (**a**) and H-plane (**b**).

**Figure 17 sensors-23-07367-f017:**
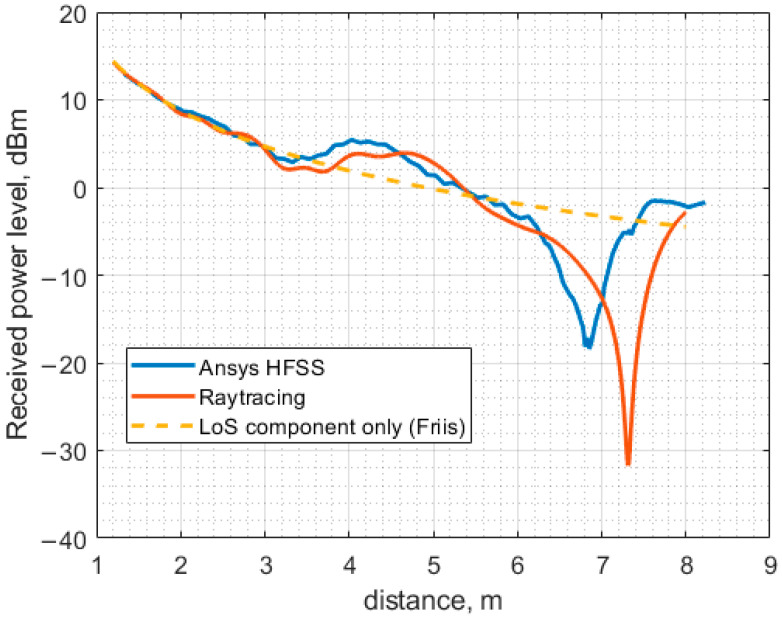
The received power level as a function of the distance between the antennas calculated using three different methods.

**Figure 18 sensors-23-07367-f018:**
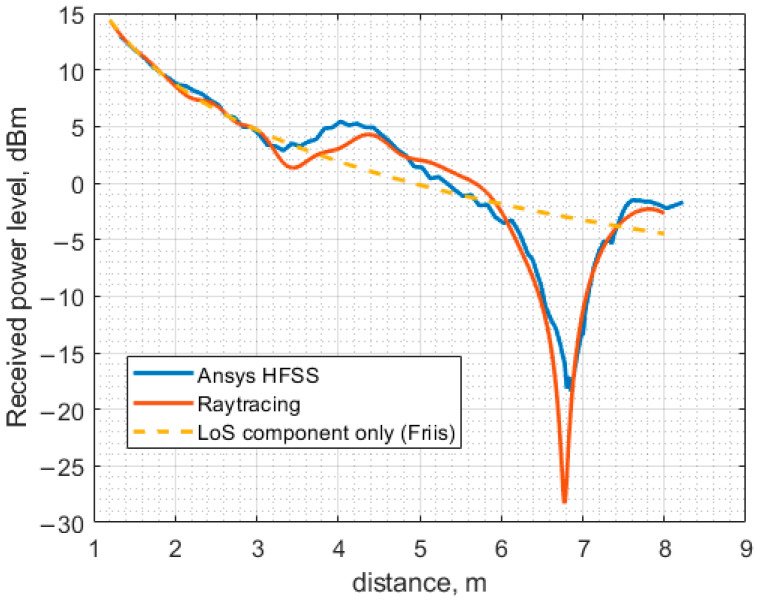
The received power level as a function of the distance between the antennas calculated using three different methods.

**Figure 19 sensors-23-07367-f019:**
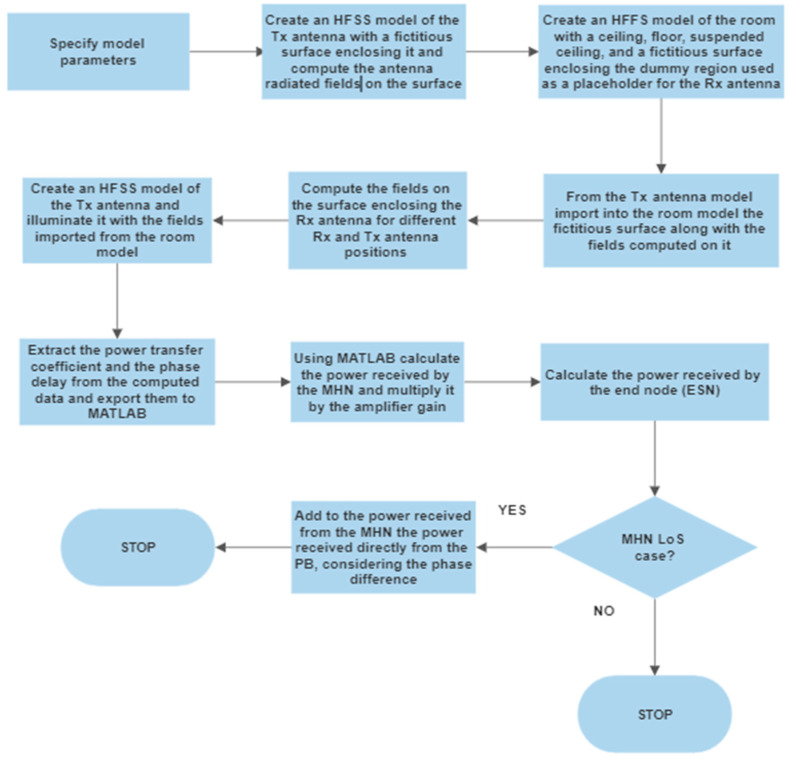
The flowchart of the transferred power estimation procedure using full-wave analysis.

**Figure 20 sensors-23-07367-f020:**
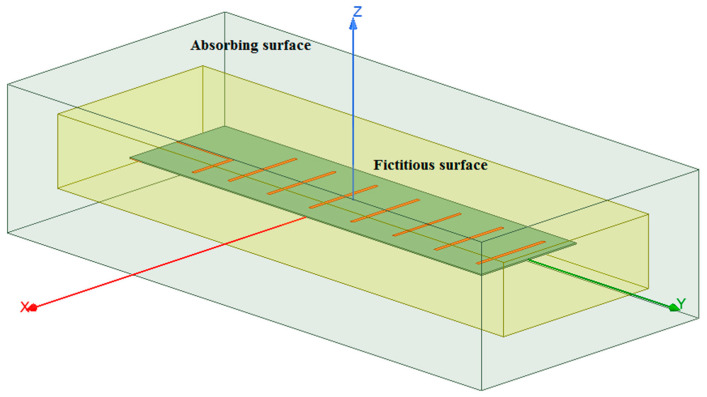
The Ansys HFSS model of a four-element Yagi–Uda antenna with a fictitious surface enclosing the antenna to compute near fields to be imported to another HFSS model.

**Figure 21 sensors-23-07367-f021:**
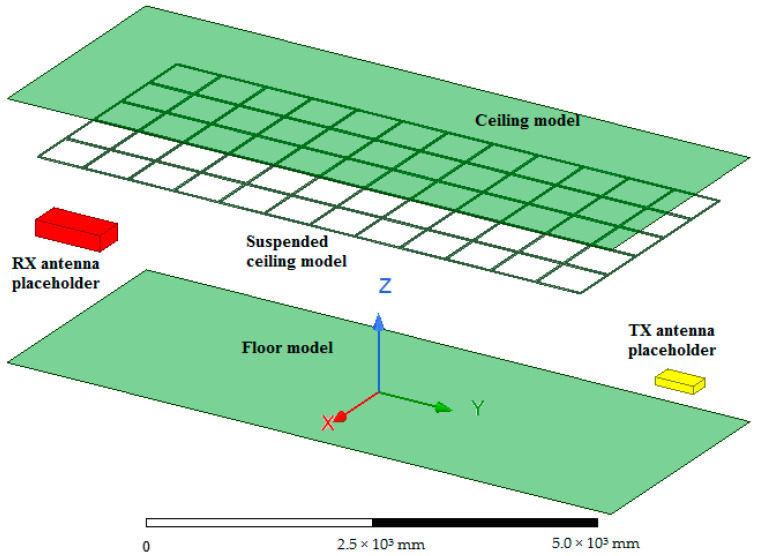
The Ansys HFSS model for the antenna-to-antenna power transfer analysis.

**Figure 22 sensors-23-07367-f022:**
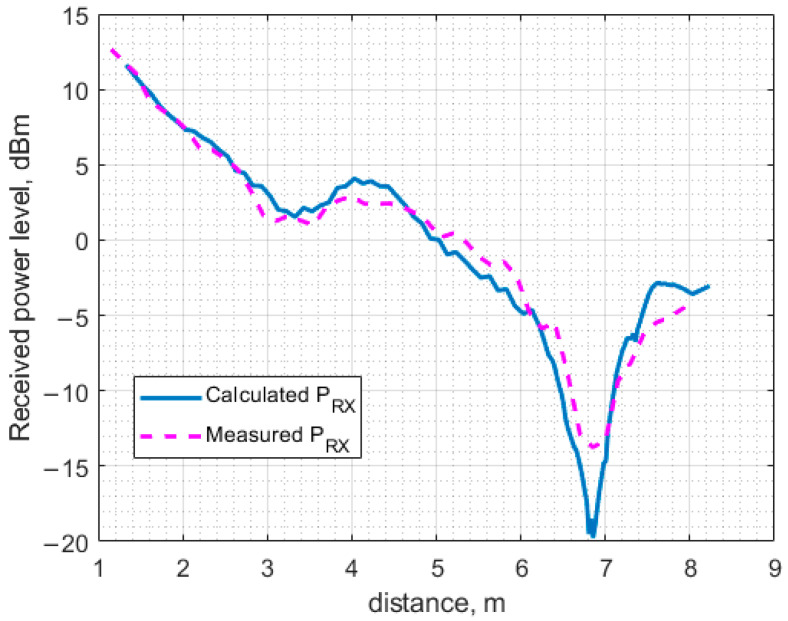
The calculated and the measured received power as a function of the distance between two four-element Yagi–Uda antennas with a fixed transmitted power of 26 dBm.

**Figure 23 sensors-23-07367-f023:**
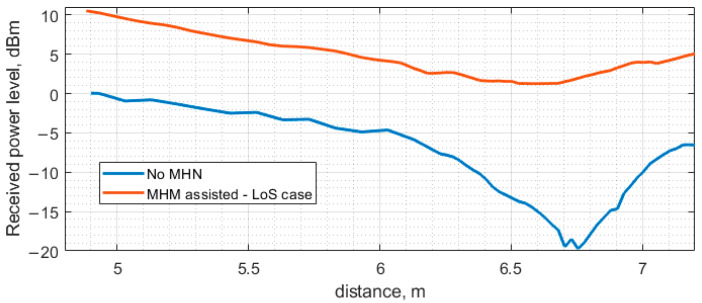
The power received at the ESN with and without the MHN when all the antennas are aligned (LoS scenario).

**Figure 24 sensors-23-07367-f024:**
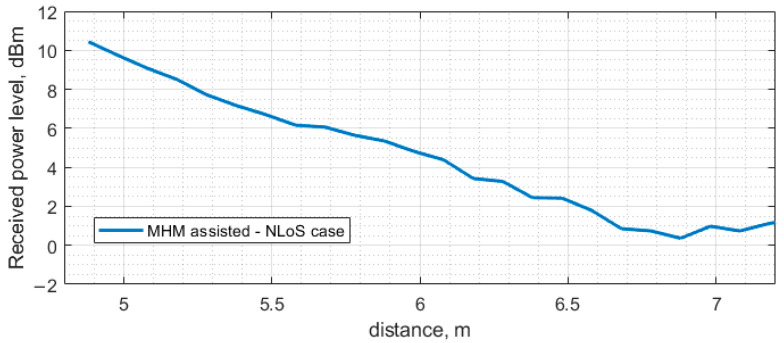
The power received at the ESN with the MHN in the case of a 90-degree turn (NLoS scenario).

**Figure 25 sensors-23-07367-f025:**
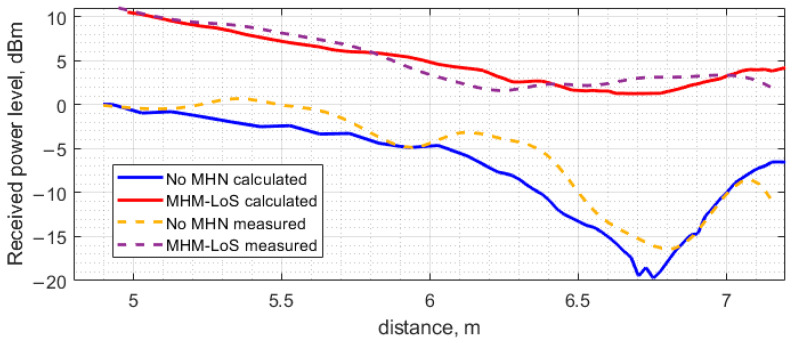
The calculated and measured power received at the ESN as a function of the distance between the PB and the ESN, obtained under the LoS scenario with and without the MHN assistance.

**Figure 26 sensors-23-07367-f026:**
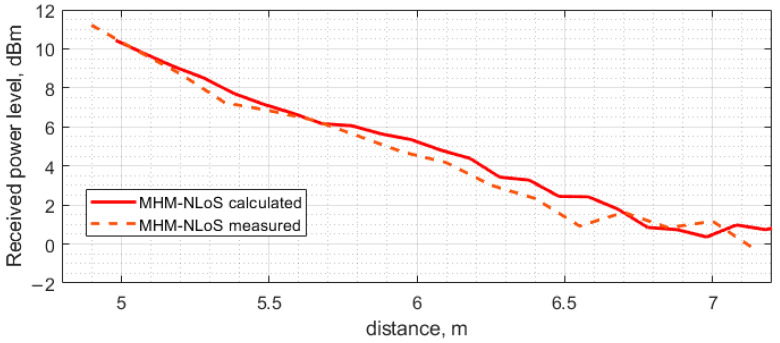
The calculated and measured power received at the ESN as a function of the distance between the PB and the ESN, obtained under the MHN-assisted NLoS scenario.

**Table 1 sensors-23-07367-t001:** Room model parameters for the ray-tracing method.

Parameter	Value
Antenna to ceiling distance	2000 mm
Antenna to floor distance	1070 mm
Antenna to the closest wall distance.	17,000 mm
Antenna to the farthest wall distance	3300 mm
Wall material (concrete) dielectric constant	4.69
Wall material (concrete) loss tangent	0.176
Floor length	∞
Floor width	∞
Ceiling length	∞
Ceiling width	∞

**Table 2 sensors-23-07367-t002:** Simplified Ansys HFSS room model parameters.

Parameter	Value
Ceiling to suspended ceiling grid distance	435 mm
Antenna to ceiling distance	2000 mm
Antenna to floor distance	1070 mm
Floor length	8000 mm
Floor width	3000 mm
Ceiling length	8000 mm
Ceiling width	3000 mm
Metal grid main (cross) tee width	590 mm
Metal grid main (cross) tee length	10 mm
Metal grid main (cross) tee height	50 mm

## Data Availability

The experimental measurement data are available in the repository https://github.com/jaancis/Wireless_Power_Transfer_MHN (accessed on 27 July 2023).

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
