# Peer review of "Efficient Multi-Hop Wireless Power Transfer for the Indoor Environment"

_sensors, 2023, doi:10.3390/s23177367_

Round 1

Reviewer 1 Report

In this study, authors present a study on Efficient Multi-Hop Wireless Power Transfer for Indoor Environment which has timely significance and lot of practical applications. The manuscript has a great potential for the publication by improving the quality of the manuscript by clarifying the following concerns of the reviewer.

11.  How does the FOM for this study, power efficiency is determined? What is the definition used?

2 2. What are the intended applications in sub-GHz range? Why particularly select this operating frequency range? What are the expected real world application with the available power level?

33. How does the proposed system is scaled up for real world practical application?

44. Impedance matching is critical aspect for maximum power transfer. How does it handle in nodes?

55. here are several elements in a one MHN. What is the typical average power consumption of a node? How does this is provided to a node?

66. Why does 865.5 MHz is selected as the operating frequency?

77. How does the accuracy of various numerical models are validated?

88. It might be better to comprehensively explain the reasons for the difference of the results in Figure 17.

99. It would also better if Mathematical FOM can be added to validate the experimental results.

110.  Placing the paper in proper scientific context in terms of real world application is expected in revised manuscript.

Appropriate and minor polishing can be done.

Author Response

The authors are grateful for the review. Here is how each point was addressed:

  1. The definition of the efficiency for the Figure 11 data was given in eq(1).
  2.  The introduction was supplemented with motivation on using the sub-GHz range - the emphasis on IoT, LORA applications and the existing WPT products by Powercast for this exact range.
  3. The conclusions section now addresses this question - the given prototype is a "proof of concept" for this type of multi-hop WPT approach, and the application for specific sensors and devices would need more fine-tuning.
  4. The description of the prototype now elaborates more on this. In short, the matching is done for the MHN charging mode, as this state is much longer than the transmission state (more details in the revised manuscript).
  5. This part is addressed in the conclusions, along with 3.
  6. This particular frequency was used due to our previous work on RF-DC converters [41]; this is the frequency we used to match the antennas we had at our disposal.
  7. The validity of the full-wave analysis results was verified by comparing them with the measured data. In addition, the accuracy of the numerical results was verified by performing simulations with different discretization element edge lengths, to ascertain the mesh parameters providing reasonable modelling accuracy. 
    Regarding the ray-tracing method, its accuracy was validated by comparing the results obtained by means of the method with those of the FEM-based full-wave analysis.
  8. This is now addressed in the revised manuscript.
  9. We assume this comment is about the data in Figure 11. At the stage of selecting the load of the ESN's RF-DC converter, the simulation/numerical study was not needed, as the goal was to practically determine the resistance that would be used.
  10. This is now covered in the introduction.

Reviewer 2 Report

Please see the attached comments.

Minor editing of English language is needed.

Author Response

The authors are grateful for the review. Here is how each point was addressed:

  1. The affiliations are done like this on the initial submission for the MDPI system to properly record the authors' information. The editors will later combine this.
  2. The abbreviations were checked and edited.
  3. The abstract was edited.
  4. The tences in the introduction were corrected.
  5. Noted and corrected.
  6. The papers R3 and R4 were used to supplement the introduction, as they were most relevant to the main topics of the introduction.
  7. The contributions are now more highlighted in the introductions section.
  8. The ratio of the ESN's harvested energy and MHN's transmitted energy is given in the results analysis.

Reviewer 3 Report

This research focused on evaluating the performance of multi-hop (MH) technology for wireless power transfer (WPT) in sub-GHz frequencies. The authors demonstrated that using a multi-hop node (MHN) can effectively compensate for path loss and increase WPT efficiency, as shown in Line-of-Sight and Non-Line-of-Sight scenarios. Comparing the results of a simplified full-wave analysis using Ansys HFSS with the ray-tracing method, the HFSS results were found to be closer to experimental data. The proposed MH WPT approach has the potential for optimizing real-life scenarios in terms of spatial distribution of MH WPT system elements. The following comments can improve the manuscript:

1-      To strengthen the introduction and emphasize the significance of IoT and WSN, consider incorporating real-world examples or statistics that demonstrate the impact and growth of these technologies.

2-      Define the term "powering issue"

3-      Clarify the motivation behind using IRSs and multi-hop routing.

4-      Since the introduction mentions that RF WPT initially branched from the topic of energy harvesting (EH), consider providing a brief explanation of EH and how it relates to the development of RF WPT.

5-      It is beneficial to review and refer to various indoor WPT applications involving moving objects such as (doi.org/10.3390/en15228643) in the second paragraph of the Introduction Section.

6-      In the context of MHN and mimicking non-direct scenarios in indoor environments, clarify the importance of the turn-on angle of 90 degrees.

7-      How does the PA's efficiency impact the overall efficiency of the MHN?

8-      What factors need to be considered to ensure seamless switching and maximize the MHN's efficiency?

9-      How was the efficiency of the RF-DC converter optimized to maximize energy harvesting from the received signal power?

10-  How does the maximum power point tracking (MPPT) function contribute to maximizing harvested power?

11-  When discussing the final MHN structure in Figure 4, consider providing a brief summary or description of each part's function (RF-DC converter, DC-DC boost converter, storage element, and two-stage amplifier).

12-  Was any specific testing or validation conducted on the individual components of the MHN prototype to measure their performance?

13-  In the NLoS scenario, where there is no direct line of sight between the PB and ESN, how was the performance of the MHN evaluated in terms of power transfer efficiency and effectiveness?

14-  How was the accuracy of the ray-tracing method-based model validated against the experimental measurement data?

15-  Consider providing a brief explanation of the specific formulation and characteristics of the Geometrical Optics approximation (GO) and Physical Optics approximation (PO) methods, as well as the Geometrical Theory of Diffraction (GTD), Physical Theory of Diffraction (PTD), and Uniform Theory of Diffraction (UTD).

The English writing of the text is generally clear and well-structured.

Author Response

The authors are grateful for the review. Here is how each point was addressed:

  1. Additional statistic on IoT and WSN growth and economic impact was added.
  2. The motivation for WPT was enhanced. The term "powering issue" refers to everything mentioned before this term. However, this was edited to avoid confusion.
  3. The motivation behind IRS and multi-hop routing was addressed: the Non-Line-of-Sight (NLoS) and WPT distance increase.
  4. Energy harvesting (EH) is a vast topic that includes vibration, heat, inductance, etc. The text was edited to emphasize that we mean the RF EH (ambient RF), and the relation between RF EH and WPT has been outlined.
  5. The paper was supplemented with relevant applications, including commercial products. As for the suggested paper, the paper is on near-field inductance-based WPT, whereas our work is on RF WPT. 
  6. The importance of the turn-on angle of 90 degrees was highlighted.
  7. This point is now addressed in the prototype's description. 
  8. This point is now addressed in the prototype's description. 
  9. This point is now addressed in the prototype's description. 
  10. This point is now addressed in the prototype's description. 
  11. This point is now addressed in the prototype's description. 
  12. Yes, the individual components were validated in our prior work [43] and the revised manuscript now refers to this.
  13.  In the NLoS case, we can only evaluate its effectiveness. The main point is that with the MHN LoS and NLoS cases, the ESN can achieve comparable power levels.
  14. The accuracy of the ray-tracing method was validated by comparing the antenna-to-antenna power transfer efficiency obtained for a single pair of antennas at different distances between the antennas with the experimentally obtained ones and that computed using Ansys HFSS.
  15. The revised manuscript now elaborates more on the methods.

Round 2

Reviewer 2 Report

The authors have basically addressed my concerns, no further comments.